# Secure IoT in the Era of Quantum Computers—Where Are the Bottlenecks?

**DOI:** 10.3390/s22072484

**Published:** 2022-03-24

**Authors:** Maximilian Schöffel, Frederik Lauer, Carl C. Rheinländer, Norbert Wehn

**Affiliations:** Microelectronic Systems Design Research Group, Department of Electrical and Computer Engineering, Technische Universität Kaiserslautern, 67663 Kaiserslautern, Germany; schoeffel@eit.uni-kl.de (M.S.); flauer@eit.uni-kl.de (F.L.); rheinlaender@eit.uni-kl.de (C.C.R.)

**Keywords:** post-quantum cryptography, low-power secure IoT, TLS, key-encapsulation mechanisms, digital signature algorithms

## Abstract

Recent progress in quantum computers severely endangers the security of widely used public-key cryptosystems and of all communication that relies on it. Thus, the US NIST is currently exploring new post-quantum cryptographic algorithms that are robust against quantum computers. Security is seen as one of the most critical issues of low-power IoT devices—even with pre-quantum public-key cryptography—since IoT devices have tight energy constraints, limited computational power and strict memory limitations. In this paper, we present, to the best of our knowledge, the first in-depth investigation of the application of potential post-quantum key encapsulation mechanisms (KEMs) and digital signature algorithms (DSAs) proposed in the related US NIST process to a state-of-the-art, TLS-based, low-power IoT infrastructure. We implemented these new KEMs and DSAs in such a representative infrastructure and measured their impact on energy consumption, latency and memory requirements during TLS handshakes on an IoT edge device. Based on our investigations, we gained the following new insights. First, we show that the main contributor to high TLS handshake latency is the higher bandwidth requirement of post-quantum primitives rather than the cryptographic computation itself. Second, we demonstrate that a smart combination of multiple DSAs yields the most energy-, latency- and memory-efficient public key infrastructures, in contrast to NIST’s goal to standardize only one algorithm. Third, we show that code-based, isogeny-based and lattice-based algorithms can be implemented on a low-power IoT edge device based on an off-the-shelf Cortex M4 microcontroller while maintaining viable battery runtimes. This is contrary to much research that claims dedicated hardware accelerators are mandatory.

## 1. Introduction

The *Internet of Things* (IoT) is disruptively changing many areas of modern life. Whether they are meant to make smart homes more comfortable, to facilitate medical- and healthcare or to enhance industrial processes in the *Industrial IoT (IIoT)*, the number of applications that utilize IoT devices is growing extensively. However, the IoT also bears numerous risks, and security is seen as one of its most critical issues, especially since the number, effort and capabilities of malicious attacks on IoT infrastructures are growing in proportion to the number of internet-connected devices. In this context, the exploit that was recently found in the *log4j*-library has shown how vulnerabilities in widely used software affect many sectors. Currently, information security is facing another threat in widely used software libraries due to the recent advances in building *Quantum Computers* (QCs).

Secure communication is essential for many IIoT applications, as the data that is exchanged via the internet may contain safety-critical, infrastructural information as well as industrial secrets [1]. In the vast majority, this security is established through cryptographic protocols such as *Transport Layer Security* (TLS) and relies on the combination of symmetric cryptography and *Public Key Cryptography* (PKC). These public key cryptosystems are hard to break on classical computers, but the theoretical basis to solve their underlying mathematical problems in polynomial time on a QC was introduced almost 30 years ago [2]. While powerful QCs were not available back then, it is expected that cryptographically relevant QCs will be available by the end of this decade [3].

Thus, the US *National Institute of Standards and Technology* (NIST) started a *Post-Quantum Cryptography* (PQC) standardization process to find new, quantum-computer-resistant public key algorithms for *digital signature algorithms* (DSAs) and *key-encapsulation mechanisms* (KEMs) [4]. During this process, candidates are evaluated based on their security assumptions and performance. Therefore, it is essential to explore the impact of key parameters such as computational complexity and bandwidth requirements of PQC algorithms in different, representative applications such as low-power, secure IoT. In addition, it is very important to know in time whether the *state-of-the-art* (SotA) low-power IoT hardware can fulfill the computational requirements of these new algorithms under the given energy constraints, or whether dedicated hardware accelerators will be required, as proposed in [5,6,7].

A holistic system view of the complete application, including the different layers of the protocol stack, is mandatory for a conclusive evaluation of the new PQC algorithms. A first investigation of post-quantum KEMs in low-power IoT devices has been carried out in [8], however, not all KEMs were included in this investigation, and some that were included have been updated recently. Furthermore, no post-quantum, secure DSA and *public-key infrastructures* (PKIs) were considered, leaving the communication susceptible to QC-based *man-in-the-middle* (MITM) attacks. In this paper, we close this knowledge gap and explore the overhead in energy demand and latency caused by the migration from conventional cryptography to PQC in a representative low-power IoT infrastructure that employs *Bluetooth low energy* (BLE), *IPv6 over low-power wireless personal area network* (6lowPAN), *Transmission Control Protocol* (TCP), TLS and *Message Queuing Telemetry Transport* (MQTT).

On the client side, we implemented post-quantum security on a low-power IoT edge device by extending mbedTLS (https://github.com/ARMmbed/mbedtls, accessed on 24 February 2022), a standard TLS library for embedded systems, to enable the seamless integration of PQC into the TLS 1.2 handshake protocol. On the server side, we updated OQS-OpenSSL (https://github.com/open-quantum-safe/openssl, accessed on 24 February 2022) with the latest Round 3 KEM versions. Additionally, we extended the library to support post-quantum DSAs and certificates. We measured the latency, energy consumption and memory requirements of post-quantum-secured TLS handshakes on the IoT edge device using the protocol stack described above and compared them to traditional handshakes based on elliptic curves.

The US NIST has announced that more performance data of PQC applications to internet protocols is necessary in the course of standardization [4]. This paper fills this gap with the following new insights:We carry out a holistic analysis of the latency and energy demand of the TLS handshake with post-quantum DSAs that are suitable for IoT and with all NIST Round 3 KEMs (“Finalists” and “Alternate Candidates” except for Classic McEliece). Our analysis shows that the communication bandwidth requirements (the sum of public key and ciphertext or signature) and memory footprint–rather than the cryptographic computations–are the major bottlenecks of PQC.We deploy new PKIs and show that, in contrast to NIST’s idea of standardizing one DSA, a combination of different post-quantum DSAs that exploits their individual benefits yields the most energy-efficient implementation for low-power IoT environments.We calculate the battery runtime for a representative IoT application with an off-the-shelf ARM Cortex M4 microcontroller and show that TLS-secured edge devices deployed with post-quantum DSAs and KEMs can achieve battery runtimes comparable to those deployed with conventional PKC.

To the best of our knowledge, this is the first work that provides a broad system view of a representative, fully post-quantum-secured, low-power IoT application with different PKIs, a representative infrastructure and including the complete protocol stack in the investigation.

This paper is structured as follows. In Section 2, we expound the background of TLS, explain how TLS can be broken with QCs and highlight the fundamentals of PQC. In Section 3, an overview of related work is provided. In Section 4, our IoT evaluation platform is presented. In Section 5, the measurement results are shown and evaluated. In Section 6, we summarize our key observations.

## 2. Background

In this section, we first explain the importance of security and, in particular, cryptography in the IoT context. The operating principles of TLS and PKI are then explained, with a focus on how they can be broken by quantum computers and the consequences of a security breach. Finally, we will show how this problem can be solved with PQC.

### 2.1. IoT Security

In the context of the IoT, security poses a challenge due to complex, application-specific infrastructures, the lack of regular in-field device updates and the limited computational power and memory resources of IoT nodes. Furthermore, the strict energy constraints of battery-powered edge devices commonly used in IoT and the requirement for a short time-to-market enforced by competitive pressure often result in drastic security risks.

This is particularly concerning since the past has shown that security breaches can lead to serious financial consequences, the loss of customer confidence and, in the worst case scenario, the loss of life. IBM found that the average cost of a data breach in 2020 was up to $7.13 million, depending on the industry sector, and for “mega breaches” IBM even estimates the cost at nearly $400 million [9]. In [10] hackers were able to gain control over essential car components, including the engine, the brake system and the steering wheel, due to the lack of authenticity checks of a firmware update mechanism which was used for malicious code insertion. In [11], hundreds of thousands of IoT devices were infected with malicious code to build botnets that were used for *distributed denial-of-service* (DDoS) or cryptocurrency mining. Although these are worst-case scenarios, the meaning of privacy, integrity and authenticity in IoT is hard to overestimate. Even lightweight sensor devices collect sensitive data when they are deployed in sensible, private environments, in industrial applications or in critical infrastructure [1]. Therefore, the usage of cryptography to provide this security should be considered essential.

### 2.2. TLS

TLS is a cryptographic protocol that protects communications against eavesdropping, tampering and message forgery and is deployed in many areas, including *Hypertext Transfer Protocol Secure* (https) [12]. PKC is utilized to provide authentication and to solve the secret-key distribution problem by an initial establishment of a master secret. This master secret is subsequently used to create the keys to encrypt the application data with symmetric cryptography, including *message authentication codes* (MACs) to ensure integrity.

The TLS standard comprises multiple protocol layers that are used during a connection [12]. This paper focuses on the TLS handshake protocol, as it uses the PKC that is susceptible to QC attacks. The handshake protocol is used to establish a cryptographically secured connection, including authentication of the communication partners, agreeing on a common master secret and the instantiation of security parameters.

Its procedure is shown in Figure 1. The relevant steps involving PKCs are explained below, as they need to be modified to be resistant to QCs. A full description of the protocol can be found in [12]. For the sake of simplicity, we only expound the high security setup that deploys ephemeral keys with mutual authentication to provide *perfect forward secrecy* (PFS).

In this setup, a DSA is used to prove the authenticity of each communication partner and to protect the communication against MITM attacks, while a key-establishment algorithm builds the pre-master secret. The procedure for an *Elliptic Curve Diffie–Hellman Ephemeral* (ECDHE) key-exchange is displayed in Figure 2 from a top-level view. Both communication partners generate a pair of keys and exchange their public keys. Then, they use their own secret key and the opposing, received public key to calculate a shared secret, which results in the same value for server and client due to the mathematical properties of ECDHE. For an attacker, it would be sufficient to have access to either s_k_C__ or s_k_S__ to calculate the same shared secret and break the cryptosystem; however, refactoring s_k_S__ from p_k_S__ requires solving the elliptic curve discrete logarithm problem, which is assumed to be intractable on conventional computers.

The protocol in Figure 2 is still susceptible to an MITM, as it can intercept the KeyExchange messages and send their own public key to client and server. Therefore, servers and clients use DSAs along with additional pairs of long-term keys to prove their identity and prevent attackers from intercepting and tampering with their messages by creating verifiable digital signatures. In the TLS handshake, the server uses its secret key belonging to DSA to sign the public key belonging to ECDHE in the ServerKeyExchange message, while the client verifies this signature with the server’s public key, what had been previously transmitted in the Certificate message as part of the server certificate. In this message, the server’s certificate chain is transmitted to establish a chain of trust containing the server’s own certificate (and optionally several intermediate certificates depending on the PKI used) and a root *certification authority* (CA) certificate (Figure 3). The client then proves its identity to the server in the CertificateVerify message by calculating the hash value of all handshake messages sent so far and signing them with its DSA. Today, the most-used DSAs are *Elliptic Curve Digital Signature Algorithm* (ECDSA) and *Rivest–Shamir–Adleman* (RSA).

### 2.3. Breaking TLS with Quantum Computers

The operating principle of QCs differs significantly from that of classical computers and is based on the usage of certain quantum mechanical effects. One of the biggest differences is the unit of information. Unlike classical computers, which use *bits* that are always in one of two distinct states (0 and 1), QCs use *qubits* that can be transformed into super-positions of states by quantum mechanical operations. This superposition is coupled with an inherent parallelism of computations in a QC, and, by employing effects such as quantum entanglement and interference with quantum logic gates, QCs can be used to efficiently solve certain mathematical problems that are practically intractable on conventional computers. Currently, QCs are not expected to be either a successor or a replacement for traditional computers but will most likely be used as hardware accelerators for them in the near future.

This heterogeneous setup also is the most appropriate one for QC-based attacks on the PKC because Shor’s algorithm, which theoretically solves the underlying problems of RSA and ECDSA (prime factorization of large integers and the discrete logarithm) in polynomial time, is comprised of two different parts [2]. In the first, classical part, the prime factoring problem is reduced to the problem of finding the order of a group, which is then solved by the QC in the second part. Although quantum computers are not yet powerful enough for these attacks, development is progressing rapidly, and it is expected that they will be able to execute successful attacks by 2031 [3].

In the context of TLS, this provides two different opportunities for attackers, as described in [8]. The first possibility for attack targets the ECDHE key exchange during the handshake and can be executed by a passive attacker by eavesdropping on one of the public keys either in ServerKeyExchange or ClientKeyExchange. Based on the public key, the attacker computes the corresponding secret key by applying the Shor algorithm and is thus able to compute the same s_k_S__ as the client and server. With s_k_S__, the attacker can decrypt all application data that is sent between server and client. This attack can already be prepared today, as encrypted data can be collected and stored with the assumption that it can be cracked with a QC in the future (“store now, decrypt later” [13]).

The second possibility targets the DSAs and can be used for an MITM attack that works as follows. The attacker eavesdrops on one of the certificates which is conveyed during a TLS handshake and retrieves the holder’s secret key with Shor’s Algorithm from the public key stored in it. Of these certificates, the root certificate offers the most extensive possibilities, as its secret key can be used to forge other certificates. Then, the attacker has all the possibilities of MITM attacks at his disposal as he/she can forge the server’s signature and use it to alter messages, thus resulting in an entire break of the TLS-implied security. For example, the attacker may provide the client with his own public key for key exchange, but signed with the server’s signature, or it may send fake commands. Both cases can have serious consequences in the IoT, especially if the device is used in critical infrastructure or in the medical sector.

### 2.4. Post-Quantum Cryptography

PQC has emerged to find new, QC-resistant PKCs due to the vulnerabilities of state-of-the-art cryptography. These new PKCs are the subject of a US NIST-conducted standardization process, which is divided into multiple rounds [14]. The new algorithms are based on different mathematical problems than conventional PKCs and based on hard problems over lattices, codes, isogenies, multivariates or hash functions that are assumed to be resistant against QCs. Currently, the process is in the last round, and the remaining candidates were divided into “finalists” and “alternate candidates”, with one of the finalists for each threatened PKC (KEMs and DSAs) to be standardized in the near future, see Figure 4. Alternate candidates are intended as a backup in case new weaknesses of finalists are detected during the process and may also be standardized in the future.

Lattice-based KEMs and DSAs are considered the prime candidates for standardization because they have fast computations and are amongst the smallest keys, ciphertexts and signatures. Most of them are based on derivations of the *learning-with-errors* (LWE) problem, which was introduced by Regev in 2005 [15] and has received considerable attention in cryptography since then. To further improve their performance, KYBER, SABER and Dilithium use more structured lattices that reduce the key size required to achieve a given level of security. This is also their main drawback, as security concerns exist regarding the extent to which this *structured-lattice-based* (SLB) construction could provide critical opportunities for algebraic attacks.

Code-based cryptography was introduced by McEliece in 1978 and is built from error-correction codes [16]. McEliece’s original approach uses a hidden, error-correcting, binary Goppa code whose generator matrix is multiplied with two randomly selected, secret matrices to build the public key, and no attack which poses a serious threat to it has been found to this day. In the NIST process, problems from coding theory are only used to construct KEMs. These code-based KEMs require larger keys than lattice-based KEMs but use very different, more mature security assumptions and are a good alternative if lattice-based KEMs become insecure.

The only candidate which is based on the super-singular isogeny walk problem is SIKE. While SIKE has clearly been shown to have very expensive calculations, e.g., in  [17], it is equipped with the lowest bandwidth requirements and provides a unique security assumption to the pool of KEMs.

Multivariate signature schemes have the smallest signatures and efficient signing and verifying operations; however, their public keys are in the range of tens of kB up to hundreds of kB. This causes very large certificate chains in TLS-like applications and limits the space for useful applications.

The hash-based DSAs have small public keys but large signatures and extensive calculations. However, SPHICNS^+^ is considered the maturest and most conservative design and, therefore, serves as a backup in case serious vulnerabilities are found in the lattice-based signature schemes.

The PQC algorithms are classified into five different security levels (see Table 1), and the above proposals contain several parameter sets for the different security levels. The minimum security requirements define how much computational resources are required to successfully break the cryptosystem; for example, attackers need at least the same resources as for a key search on *Advanced Encryption Standard-128* (AES-128) to break NIST level 1 KEMs [14].

## 3. Related Work

This paper combines the application of post-quantum cryptography to servers and embedded devices, its integration into existing protocols and energy-optimized secure IoT infrastructures. Because these areas are themselves still the subject of research, the amount of directly related work is very small.

On the server side, *Open Quantum Safe* (OQS) is one of the most important projects that seeks to simplify prototyping of PQC on standard computers and servers [18]. In this project, PQC algorithms have been integrated into OpenSSL implementations (https://github.com/open-quantum-safe/openssl, accessed on 24 February 2022) for TLS 1.2 and TLS 1.3 versions. In  [19], a network emulation framework was developed using the OQS implementations to evaluate the advantages and disadvantages of PQC procedures in TLS. They showed that in fast, reliable networks, TLS handshake completion time is mainly determined by PKC computation, while in networks with higher packet loss rates, communication has a stronger impact on completion time.

For the client side, the pqm4 project developed a library with assembler-optimized versions of most PQC primitives for the ARM Cortex M4 processors that are also used in this work [20]. Kannwischer et al. performed benchmark tests with this library and showed that significant speed-ups can be achieved for most primitives through their optimization. In  [21], the energy demand of the calculations with the same library were measured, and it was shown that some of the lattice-based algorithms have the lowest energy demand out of all NIST proposals.

A fully post-quantum TLS implementation on embedded devices was presented in [22]. The authors integrated one parameter set of the post-quantum KEM KYBER and the DSA SPHINCS+ into the mbedTLS library and conducted measurements on three different devices, where the devices acted either as server or client. They concluded that TLS handshakes with those two schemes are feasible on their devices; however, they omitted the impact of the communication overhead in their measurements, observed only one KEM and one DSA and did not provide information on the energy demand. A first observation of some post-quantum KEMs for low-power IoT devices has been carried out in [8], and it has been shown that dedicated hardware accelerators are not mandatory for an efficient implementation. However, the observation did not include all NIST Round 3 KEMs, and it is unexplored whether the same observations hold for post-quantum DSAs.

To date, research has focused solely on the cryptographic computations on embedded systems or analyzed standard desktop applications. To the best of our knowledge, no work has provided a holistic system view of the performance of post-quantum DSAs and KEMs in a representative low-power IoT application. This is somewhat surprising when considering the large amount of research on energy-optimized hardware accelerators for PQC algorithms, e.g., [5,6,7], where it is claimed that hardware accelerators are necessary to facilitate the usage of PQC on embedded devices. We close this knowledge gap in this paper by presenting an in-depth investigation of post-quantum DSAs and KEMs in a low-power IoT system context. With this investigation, we support NIST’s call for more performance data of PQC in protocol-based applications [4].

## 4. Post-Quantum Safe IoT Infrastructure

This section describes the IoT infrastructure used in this paper as a setup to evaluate the candidates that are considered for standardization by the US NIST. Furthermore, we propose feasible PKIs for this test setup and explain which KEMs and DSAs were investigated.

### 4.1. Low-Power IoT Evaluation System

IoT infrastructures consist of IoT nodes (actors or sensors), gateways and the cloud server. In this paper, we deployed a typical secure, low-power IoT infrastructure based on [23], see Figure 5.

The nRF52840 *system on a chip* (SoC) by Nordic Semiconductor was selected for the battery-powered edge devices because it uses the ARM Cortex M4 *micro controller unit* (MCU), which was chosen by the U.S. NIST as a reference platform for embedded systems through the PQC process [24]. Furthermore, the SoC has been shown to meet the strict energy requirements of low-power IoT environments in previous work [23]. The SoC includes a security subsystem (ARM CryptoCell-310), a Bluetooth 5-compatible 2.4 GHz radio, 1 MB of flash memory and 256 KB of *random-access memory* (RAM). The ARM CryptoCell provides hardware acceleration for *elliptic-curve DSA* (ECDSA) and ECDHE key exchange and also has hardware accelerators for AES128 and *Secure Hash Algorithm 2* (SHA2). The standard TLS protocol, rather than variants such as *Datagram Transport Layer Security* (DTLS), is used together with TCP and IPv6 to provide true end-to-end encryption between the edge device and the server.

The server is a standard Linux server with a MQTT broker and an Intel Xeon Silver 4214 @ 2.2 GHz, which features the *Advanced Vector Extension* (AVX) instructions. These instructions are used by many authors of the PQC algorithms in their optimized implementations and therefore facilitate the mitigation to PQC in our setup.

A Raspberry Pi 3 is used as the gateway, and, since both communication partners use the standard TLS, there is no need for time-consuming decryption and re-encryption on it. Therefore, the gateway acts as a headless router and forwards the communication from the BLE physical layer to the Ethernet physical layer and vice versa without affecting the TLS layer. For a trade-off between connection handling, data throughput and energy demand, a BLE connection interval value of *20ms* has been chosen [23].

Even though TLS 1.3 is the most recent version, TLS 1.2 is deployed in this setup for the following reasons. First, TLS 1.2 is still widely used, particularly in the embedded environment. A reason for this is that the library mbedTLS (https://github.com/ARMmbed/mbedtls, accessed on 24 February 2022), which is quite popular because of its very unrestricted licence, does not yet officially support TLS 1.3. Second, one of the advantages of TLS 1.3 handshakes over TLS 1.2 handshakes is achieved by reducing the number of roundtrips, which has a much smaller impact on the latency in our setup, given PQC’s relatively large keys.

The NIST curve p-256 is used for all elliptic-curve-cryptography-based operations in this setup, as a ECDHE_ECDSA connection is used to compare classical with post-quantum cryptography. Analogous to conventional ECDHE-based handshakes, ephemeral keys are generated on every new handshake with PQC to ensure PFS.

The power consumption and the latency of the relevant events in the TLS handshake were recorded by a Power Profiler Kit 2 from Nordic Semiconductor and double-checked by a high-resolution Keithley DMM7510 digital multimeter.

### 4.2. PQC-Based Public Key Infrastructure

A low complexity PKI was deployed in the IoT setup, see Figure 6. The server and IoT node certificates are both signed by the same trusted root CA. No intermediate certificate was used in order to limit the necessary amount of data for conveying the certificate chain during the TLS handshake.

Besides metadata such as expiration date, subject name and organization, it is primarily the public keys and the signature of the DSA that determine the certificate size for PQC. In the given setup, the server sends its certificate chain (root certificate and the server certificate) to the client, while the client transmits its own certificate. Thus, the required bandwidth (bandwidth normally denotes the rate of data transfer—in the context of PQC, we denote bandwidth requirements as the amount of data that must be exchanged between server and client (public key and ciphertext in the case of KEMs and public key and signatures in the case of DSAs))in the TLS handshake with respect to the DSAs $bw_{DSA}$ is
(1)$$bw_{DSA}=|p_{kC}|+|sig_{C}\left|+\right|p_{kS}|+|sig_{S}\left|+\right|p_{kCA}|+3\cdot |sig_{CA}|$$
where $|p_{kC}|$ denotes the byte-size of the client’s public key, $|sig_{C}|$ denotes the size of the client’s signature and the indices *S* and *CA* the same for server and *CA*, respectively. In conclusion, the signature size of the root *CA* has a large impact on the required bandwidth.

Table 2 shows the six remaining DSAs in the standardization process with their key and signature sizes in comparison to conventional state-of-the-art signature algorithms. The table shows that the key and signature sizes of PQC methods are several orders of magnitude larger than those of conventional methods. Furthermore, there are also very large differences within these new algorithms. While the lattice-based Dilithium and Falcon have moderate sizes, Rainbow, GeMSS, and Picnic have high bandwidth requirements, and a single certificate based on any of them would already exceed the maximum TLS handshake message size of 2^14^ bytes. A chain of two SPHINCS^+^-based certificate would also exceed this boundary, and SPHINCS^+^ has also been shown to have computational intensive sign and verify operations [17]. Therefore, we only consider Dilithium and Falcon as feasible in a typical IoT setup and did not further investigate other methods in this paper.

The US NIST plans to standardize only one out of these two algorithms [31]. Nonetheless, they have their individual benefits regarding security, bandwidth requirements and computational complexity of the sign and verify operations. Hence, we also deploy PKIs where the client authenticates itself by using Falcon and the server by using Dilithium and vice versa to determine the most energy- and latency-efficient solution for our IoT setup.

Furthermore, in many applications, it may not even be necessary to make the entire PKI quantum computer secure. Applying quantum computing will still be very expensive, especially in the early days, and so attackers will probably increasingly go against much-used certificates, such as the root certificate or the server, rather than attacking individual client certificates. Therefore, we propose a heterogeneous PKI where the client authenticates itself with classical ECDSA, whose public key is content of a post-quantum signed certificate, while the server and root CA only use PQC.

Unlike DSAs, KEMs only require a single key and ciphertext to be exchanged, and they are not both part of the same message either. On top of that, most candidates have moderate key sizes, see Table 3. Therefore, the impact of their bandwidth requirements is not as large as that of DSAs, and all of the KEMs are theoretically feasible for a typical IoT setup at least in the level 1 parameter set. This excludes Classic McEliece, which has extensively large public keys that exceed the RAM size of standard MCUs.

### 4.3. Software Implementation

TLS-based security with post-quantum key exchanges and DSAs was implemented on server and edge devices. No adjustments were required on the gateway compared to a conventional TLS-based setup as it only interfaces in the communication on the physical layer.

#### 4.3.1. Edge Devices

Although mbedTLS only supports TLS 1.2 instead of 1.3, it was selected over wolfSSL for implementing TLS on the edge device due to its less restrictive license. The already modified mbedTLS version from [8] was extended to support all NIST Round 3 KEMs and the above selected DSAs in TLS handshakes. To support the above described PKI variations, the functionality to create post-quantum secure certificates as well as classical certificates that were signed by a post-quantum algorithm were implemented. The open-source library *Lightweight IP* (lwIP) was used to provide the TCP/IP protocol stack.

The NIST submissions include a C-reference implementation of each PQC algorithm. Furthermore, the pqm4 project [17] optimized most algorithms for the Cortex M4. In this paper, the pqm4 versions were used for all KEMs except BIKE and HQC. However, for the DSAs, the statically allocated portions of RAM in the pqm4 library were too large for the given setup. Therefore, the NIST reference implementations were used for them.

Similar to [8], we used the ARM Cryptocell for the AES128 and SHA2 calculations inside the PQC algorithms, where they served as building blocks for *extendable output functions* (XOF). SHA3 has been implemented by the Cortex M4 assembler-optimized version of the Keccak Team (https://github.com/crystalsnetworkdev/pq4, accessed on 24 February 2022).

#### 4.3.2. Server

The open-source Eclipse Mosquitto is used as MQTT broker (https://github.com/eclipse/mosquitto, accessed on 24 February 2022) on the server, which provides an *application programming interface* (API) for the integration of OpenSSL. The TLS 1.2 version of the OQS project, OQS-OpenSSL (https://github.com/open-quantum-safe/openssl, accessed on 24 February 2022), was used to implement post-quantum security on the server. As support for TLS 1.2 was dropped by the project in 2020, we updated the library with the newest KEM versions. Furthermore, the library does not support post-quantum DSAs and certificates. Therefore, we implemented these features.

## 5. Results and Discussion

This section covers the measurement results and their discussion. First, we measured the latency and energy consumption of TLS handshakes with post-quantum KEMs and compared them to conventional ECDHE-based handshakes. The overhead that was caused by deploying KEMs with higher security levels was also investigated. Then, the latency and energy consumption of different PKIs were examined to identify the individual advantages and disadvantages of the DSAs. Based on these results, we calculated theoretical values for the battery runtime of edge devices for the different KEMs and PKIs.

All measurements were carried out ten times on the client side for each algorithm-respective PKI. We recorded the TLS handshaking procedure, starting with sending the ClientHello message on the edge device and ending with receiving the Finished message from the server. The figures show the mean values of the measurements, and the standard deviation is shown in the graphs with error bars. The exact numerical values of the measurement results are shown in Table A1 and Table A2 in the Appendix A.

### 5.1. KEMs

We integrated the NIST Round 3 KEMs into the OQS-OpenSSL library on the server and mbedTLS library on the client to investigate their latency, energy consumption and memory footprint on the edge device. During the measurements, the previously introduced PKI is used with conventional ECDSA certificates.

Figure 7 shows the TLS 1.2 handshake latencies with all NIST Round 3 KEMs except Classic McEliece. The latency of a handshake with conventional, hardware-accelerated ECDHE is included for comparison. We used this latency subtracted by the computation time of ECDHE as a baseline for the remaining measurements (blue bar in the figure). The time for performing the cryptographic calculations for the KEM (Encapsulation) or ECDHE on the client is represented by the orange bar. The yellow bar represents the total handshake time subtracted by the orange and blue bars. This corresponds to the communication overhead that is caused by the larger key sizes of PQC primitives compared to the conventional ECDHE. This bar also includes the server’s computation time for the new algorithms, which, however, had a marginal share.

The figure shows that the cryptographic calculations of most KEMs account for a very small fraction of the total handshake latency. This observation is contrary to the investigations of conventional handshakes, where it was shown that the (software-based) computation of ECDHE has a significant impact on latency [1]. In contrast, as shown by the yellow bar, the larger public keys and ciphertexts of the KEMs cause a medium to large increase in handshake latency. The finalists, KYBER, NTRU and SABER, and the alternate candidate, NTRU-Prime, are the best performing ones and require roughly 25% more time than ECDHE in the level 1 parameter set. Even with the high security parameter-sets of those KEMs, Encaps has a low percentage of total handshake latency, which is about twice that of ECDHE.

The corresponding energy demand is shown with a logarithmic scale in Figure 8. Handshakes with the lattice-based finalists and NTRU-Prime require energy comparable to those with ECDHE. This observation also holds for higher security levels of the lattice-based KEMs. SIKE, however, requires significantly more energy than the rest of the methods due to its complex calculations, especially at higher security levels.

Figure 9 displays the maximum stack usage of the KEMs during the Encaps operation in our setup. As presented, KYBER512 (2.5KB) and SIKE_P434 (4KB) have the smallest memory footprints, which is well-suited even for devices with low memory resources. In contrast, the code-based KEMs have the largest memory footprints. This is also the reason why we only implemented the level 1 parameter-set. On top of the stack memory requirements, some of the KEMs also increase the required size of statically allocated input/output buffers across the protocol stack of mbedTLS and lwIP. While conventional PKC has little influence on those buffer sizes, PQC methods with large keys, such as FrodoKEM, require an adjustment of those buffers across multiple layers, thus further increasing the penalty of such large keys.

In summary, all of the lattice-based KEMs that are still part of the NIST process are feasible in a low-power IoT device. The benefit of using dedicated hardware accelerators is very limited, as their latency is mostly determined by the bandwidth requirements rather than the computation in our setup. The alternate candidate, SIKE, has a high energy demand in all parameter sets due to its computational complexity; however, with a sophisticated hardware accelerator it could surpass the lattice-based KEMs due to its smaller keys and ciphertexts. In our setup, KYBER512 performs the best due to its low stack usage. We, therefore, selected it for the following investigations of DSAs.

### 5.2. DSAs

In accordance with US NIST’s goals of standardizing only one method, we initially used PKIs based on a single signature method to evaluate the overhead of pq-DSAs. However, unlike KEMs where only one encapsulation operation is performed on the client, DSAs are used multiple times during the handshake. Therefore, they offer higher degrees of freedom in terms of their integration into the setup. As proposed in Section 5, we also implemented heterogeneous PKIs that use a combination of classical and post-quantum DSAs, as well as PKIs that use both Falcon and Dilithium.

The TLS handshake latencies based on PKI are displayed in Figure 10. The blue bar represents the time of a KYBER512-based handshake with a fully ECDSA-based PKI. This corresponds to the handshake latency measured in Figure 7 and was used as a baseline. The green bar indicates the time needed to verify the server’s signature over the public key and the CA signature over the server’s certificate. The orange bar shows the time during which the client performs the signing operation. Similar to the previous figure, the yellow bar represents the communication overhead caused by the migration from classical to pq-DSAs due to the larger key sizes, signature sizes and certificates. The corresponding energy consumption is displayed in Figure 11.

The chart shows that, in contrast to the KEMs, all post-quantum DSAs cause a significant increase in TLS handshake latency. PKIs that deploy Dilithium2-based server, client and CA certificates show the largest latency (second bar from the left), which is about four times that of the conventional ECDSA-based handshakes. Consistent with previous observations, this is mainly caused by the large communication overhead due to the high bandwidth requirements of Dilithium. This is different from the pure Falcon512-based PKI (sixth bar from the left), as it reduces latency by half compared to the Dilithium2 version. However, the signing function of Falcon512 is one of the main bottlenecks, as it has a major influence on latency and energy demand.

We combined Dilithium2-based client certificates with Falcon512-based server and CA certificates to mitigate these disadvantages (fifth bar from the left). The charts show that the latency is slightly higher compared to the pure Falcon512 solution, but the energy demand is reduced by 30%. The lowest energy requirements and latency are achieved with the combination of a traditional ECDSA-based client certificate and Falcon512-based server/CA certificates, reducing the overhead of migrating to such new PKIs from 230% to 30% in energy and from 115% to 50% in latency compared to purely ECDSA-based PKIs.

In summary, post-quantum DSAs cause a strong increase in latency and energy demand in low-power IoT devices if deployed in homogeneous PKIs as planned by the US NIST. This is caused by the fact that Falcon has a computationally intense signing function and Dilithium has efficient computations but large bandwidth requirements. However, an efficient PKI can be built either by combining Dilithium-based client certificates and Falcon-based server and CA certificates or by deploying a dedicated hardware accelerator for the Falcon signing operation on the client side. To increase the efficiency further, conventional ECDSA-based certificates can be used on the client side to protect it at least against classical attacks, which is likely to be a high enough barrier in most applications due to the expected limited access to quantum computers in the near future.

### 5.3. Battery Life Analysis

We used a representative, standard, lithium cell (1/2 AA, thionyl chloride, 3.6 V, 1.2 Ah) as the energy source, and a sleep-current of 2.5 µA was measured to analyze the battery run time for different PKIs and KEMs. To compensate for aging effects, 70% of the stored energy was assumed to be usable. In contrast to the previous results, we measured not only the TLS handshake, but a full TLS-secured MQTT transaction in which 12 bytes of payload data, which could come, for example, from a sensor, were transmitted in encrypted form. Since the length of the payload is highly application specific and in this work we focused on the overhead of the mitigation to post-quantum secured handshakes, the payload was kept small to facilitate the investigation of the overhead of these new handshakes. Energy for sensor data acquisition was not included in the estimations, as it strongly depends on the deployed sensor.

Figure 12 shows the runtime estimation for different KEMs and PKIs depending on different communication counts per day. The KEMs are labeled with their corresponding name in the legend, the different PKIs are denoted with cliDSA_srvDSA where cliDSA denotes the DSA of the client certificate and srvDSA denotes the DSA of the server certificate. The chart shows that the theoretically achievable battery runtime of the KEMs differs only slightly for many algorithms, including the conventional ECDHE. KYBER512 with ECDSA client certificate and Falcon512 server and CA certificates can still achieve 10 years of battery life with more than 200 data transfers per day. Even though a fully Dilithium2-based PKI induces a relevant overhead compared to conventional cryptography, the estimation shows that this overhead is still very low with respect to battery life time.

Although the battery runtime is an estimation, it shows that post-quantum, TLS-secured communication is feasible with state-of-the-art hardware, leaving sufficient energy for sensor data collection or further applications. Furthermore, we observed that battery runtime is not significantly affected by migrating from conventional cryptography to PQC.

## 6. Conclusions

In this work, we conducted an in-depth investigation on the application of new PQC algorithms to TLS-secured, low-power IoT devices. The TLS libraries OpenSSL and mbedTLS were extended to support the integration of NIST Round 3 KEMs, DSAs and post-quantum safe certificates that are suitable for IoT TLS handshakes. The energy consumption, memory footprint and latency of TLS handshakes with quantum-computer-resistant KEMs, DSAs and PKIs were explored and evaluated in a representative low-power IoT infrastructure to identify the trade-offs of the different algorithms. Based on our investigation, we provided estimations for the lifetime of battery-powered IoT edge devices that employ quantum-computer-resistant communication. Our key observations are:**A holistic system view is necessary:** Most research focuses on the computational complexity of PQC without considering other parameters, such as the associated communication overhead. However, a conclusive evaluation requires a broad system view that encompasses the entire application in typical environments to determine the advantages and disadvantages of the individual algorithms.**PQC is feasible with SoTA hardware:** Efficient implementations of post-quantum-safe TLS are possible with off-the-shelf hardware on IoT edge devices, as the energy consumption of PQC leaves sufficient energy for sensor data collection or other applications while still maintaining viable battery runtimes. This is contrary to many publications that claim that dedicated hardware accelerators are required to implement PQC on resource-constrained devices, without providing conclusive data to back this assumption up.**Key and signature sizes are the major bottleneck:** The energy and latency overhead of most DSAs and KEMs in our IoT setup is primarily driven by the larger bandwidth requirements rather than the computational overhead. In addition to the previous observation, the potential energy and latency benefits of dedicated hardware accelerators are very limited for low-power IoT applications, especially for the popular lattice-based KEMs and the DSA Dilithium, for which accelerators have been proposed in many publications. The only exceptions to this are applications where Falcon is used for client-side signing or SIKE, as then latency and power requirements are significantly affected by computations.**Heterogeneous PKIs reduce latency and energy consumption:** A combination of different post-quantum DSAs that exploits their individual benefits regarding bandwidth requirements and computational complexity yields the most energy-efficient implementation for low-power IoT environments. Our results show that PKIs combining Dilithium-based client certificates and Falcon-based server and CA certificates together with KYBER-based key-exchanges achieve the best trade-off between bandwidth requirements, latency and energy consumption. This is contrary to NIST’s idea of standardizing only one procedure.**Classical and PQC-based PKIs optimize the trade-off between security and efficiency:** In many applications, not all communication partners in the PKI need to utilize post-quantum-safe certificates. If the client deploys conventional ECDSA as the signing algorithm, the drawbacks of post-quantum-secure PKIs are greatly reduced while maintaining a reasonable attack barrier for the near future.

These results prove that the usage of post-quantum KEMs and DSAs is already feasible today in terms of energy demand and latency for low-power IoT devices with state-of-the-art hardware. Nevertheless, the overhead of PQC, especially DSAs, can be significantly reduced by decreasing the size of signatures and public keys rather than simplifying the computations, and we would like to motivate further research in this direction.

## Figures and Tables

**Figure 1 sensors-22-02484-f001:**
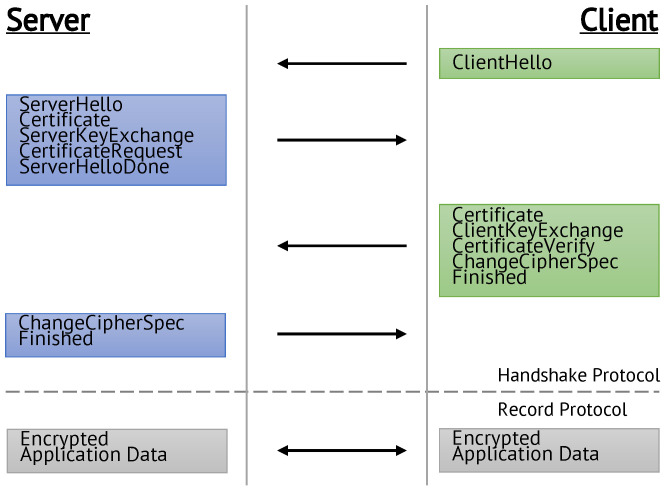
Procedure of the TLS 1.2 handshake as defined in [12].

**Figure 2 sensors-22-02484-f002:**
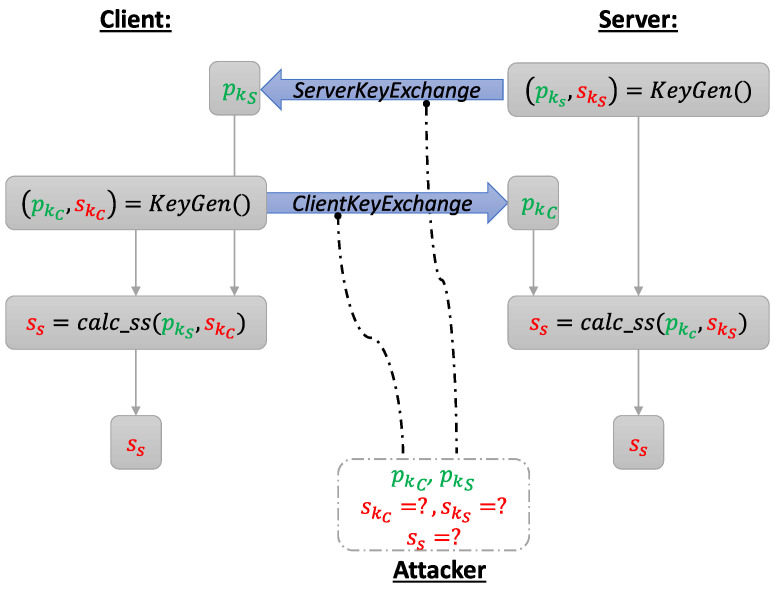
ECDHE working principal in the TLS 1.2 handshake: respectively, p_k_C__ and p_k_S__ denote the public key of client and server, s_k_C__ and s_k_S__ the secret key of client and server, and s_s_ the shared secret. The green-marked variables may be publicly known while the red ones have to remain secret.

**Figure 3 sensors-22-02484-f003:**
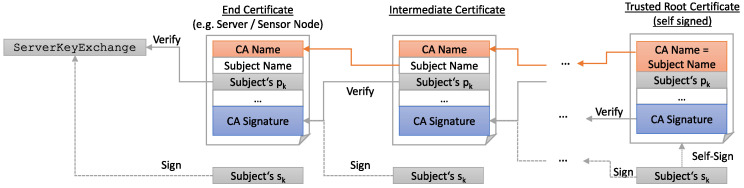
Example for a certificate chain and its application to the TLS handshake.

**Figure 4 sensors-22-02484-f004:**
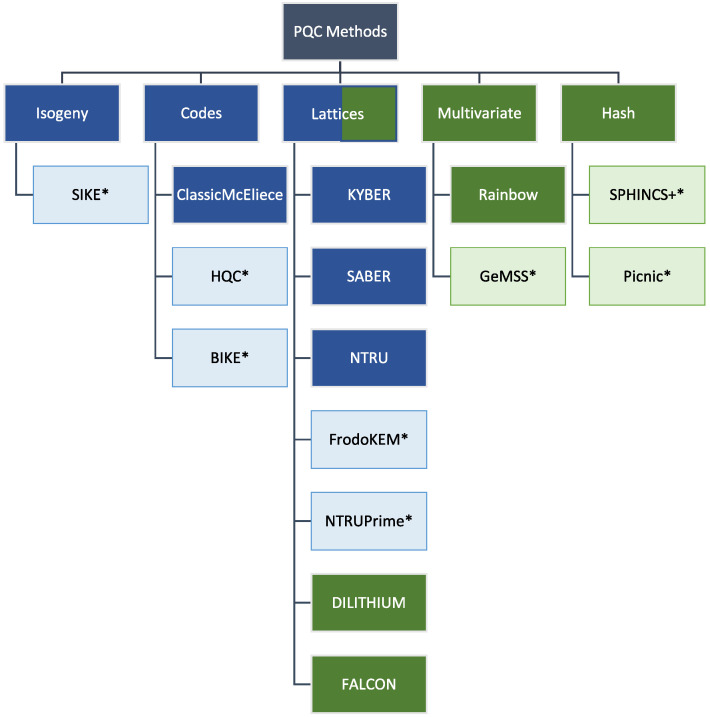
PQC algorithms that are currently considered for standardization in the associated NIST process. KEMs are marked with blue-colored boxes and DSAs with green-colored boxes. Asterisks and lower opacity indicate alternate candidates.

**Figure 5 sensors-22-02484-f005:**
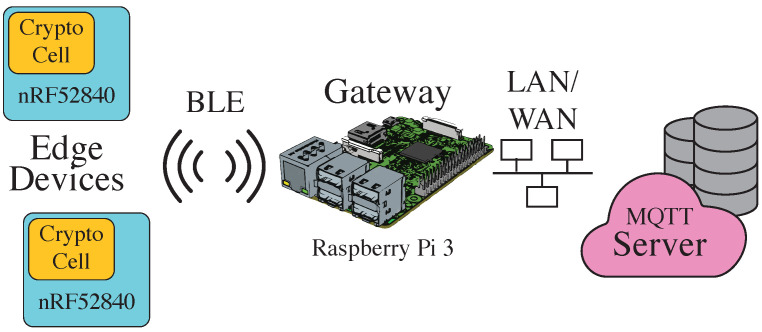
Network topology of the evaluated IoT system [23].

**Figure 6 sensors-22-02484-f006:**
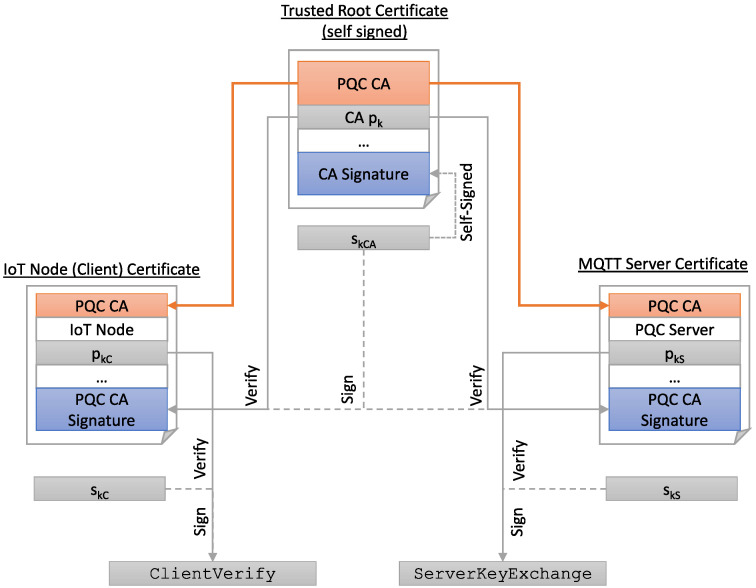
PKI that is used in the measurement setup.

**Figure 7 sensors-22-02484-f007:**
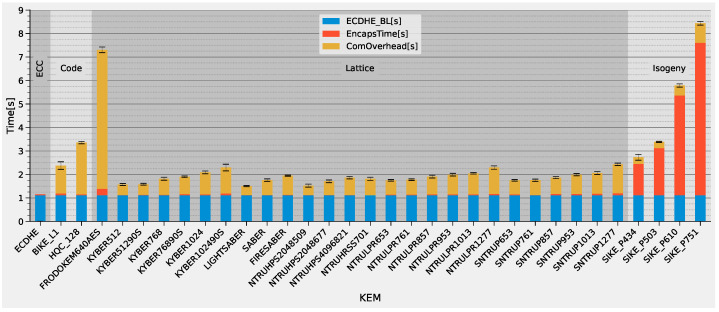
TLS 1.2 handshake latency for all NIST Round 3 KEMs with 20 ms BLE connection interval.

**Figure 8 sensors-22-02484-f008:**
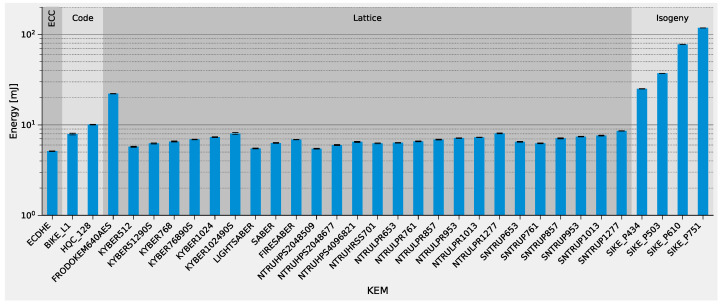
TLS 1.2 handshake energy for all NIST Round 3 KEMs with 20ms BLE connection interval.

**Figure 9 sensors-22-02484-f009:**
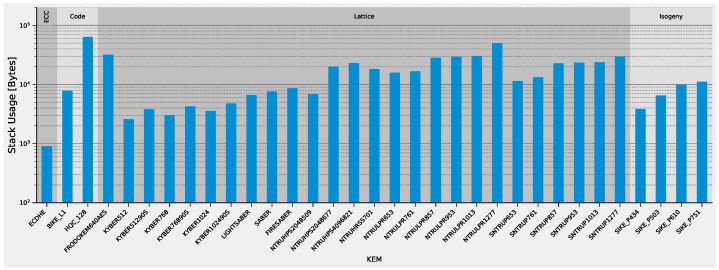
Stack usage of the encapsulation operation of the NIST Round 3 KEMs.

**Figure 10 sensors-22-02484-f010:**
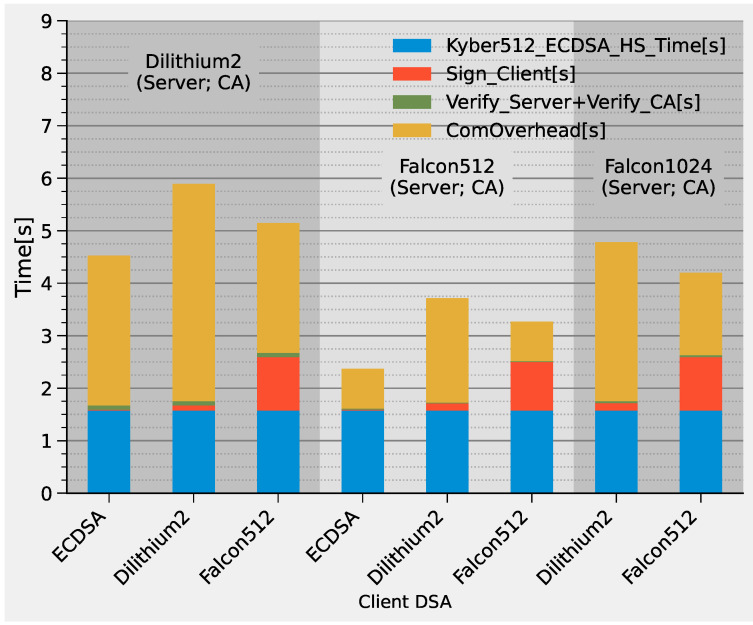
TLS 1.2 handshake latency for different PKIs with post-quantum DSAs.

**Figure 11 sensors-22-02484-f011:**
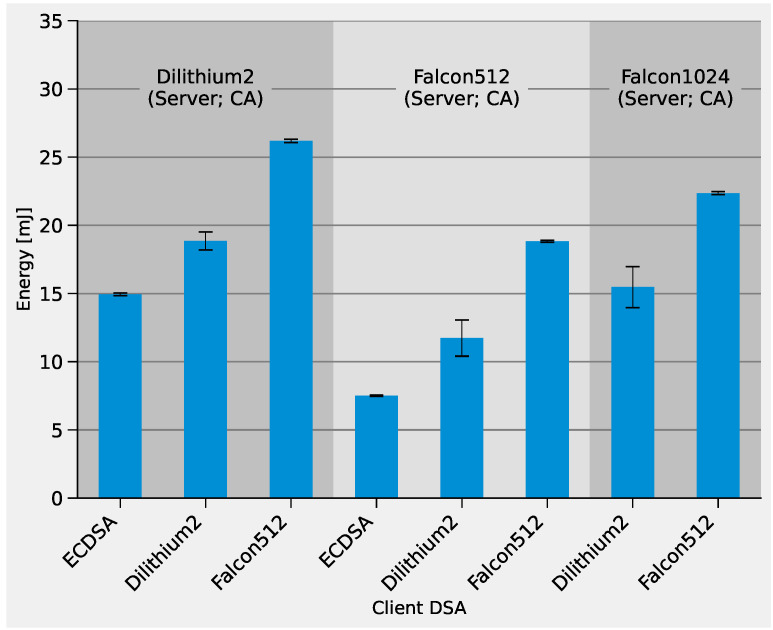
TLS 1.2 handshake energy demand for different PKIs with post-quantum DSAs.

**Figure 12 sensors-22-02484-f012:**
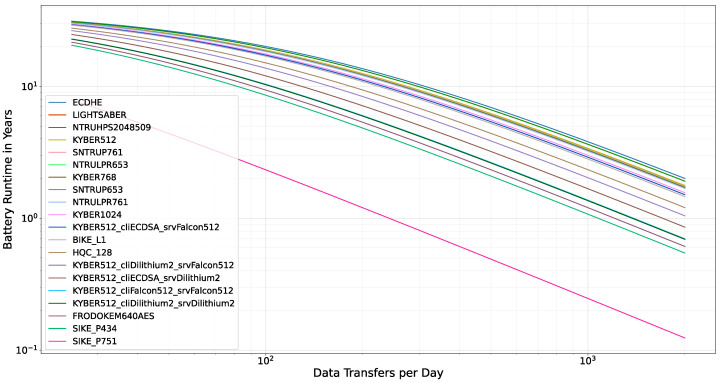
Battery runtime estimations for different KEMs.

**Table 1 sensors-22-02484-t001:** NIST security levels in the PQC standardization process as defined in [14].

Security Level	Minimum Security Requirements
1	AES-128
2	SHA-256
3	AES-192
4	SHA-384
5	AES-256

**Table 2 sensors-22-02484-t002:** US NIST Round 3 DSAs. The NIST Level shows the NIST security level according to Table 1; $|p_{k}|$ and |sig| denote the size of the public key and signature in bytes. We also included the state-of-the-art DSAs RSA and ECDSA for comparison.

Scheme Name	Competition Status	Problem Family	Parameter-Set	NIST Level	$|p_{k}|$	|sig|
			dilithium2	2	1312	2420
**Dilithium** [25]	finalist	lattice	dilithium3	3	1952	3293
			dilithium5	5	2592	4595
**Falcon** [26]	finalist	lattice	falcon512	1	897	690
			falcon1024	5	1793	1330
			I_Compressed	1	58.8 k	66
**Rainbow** [27]	finalist	multivariate	III_Compressed	3	258.4 k	164
			V_Compressed	5	523.6 k	212
			WhiteGeMSS128	1	358.2 k	30
**GeMSS** [28]	alternate	multivariate	WhiteGeMSS192	3	1294 k	47
			WhiteGeMSS256	5	3222 k	64
			picnic3-L1	1	34	13,802
**Picnic** [29]	alternate	hash	picnic3-L3	3	48	29,750
			picnic3-L5	5	64	54,732
			sphincs-128s	1	32	7856
**SPHINCS^+^** [30]	alternate	hash	sphincs-192s	3	48	16,224
			sphincs-256s	5	64	29,792
**Conventional Crypto**	(None)	ECDSA	sec256r1	1	65	71
		RSA	RSA-2048	1	256	256

**Table 3 sensors-22-02484-t003:** US NIST Round 3 KEMs. The NIST Level shows the NIST security level according to Table 1; $|p_{k}|$ and |ct| denote the size of the public key and ciphertext in bytes. We also included the state-of-the-art key-exchange algorithms RSA and ECDHE for comparison.

Scheme Name	Competition Status	Problem Family	Parameter-Set	NIST Level	$|p_{k}|$	|ct|
			mceliece348864	1	255 K	128
			mceliece460896	3	512 K	188
**McEliece** [32]	finalist	codes	mceliece6688128	3	1 M	240
			mceliece6960119	5	1 M	226
			mceliece8192128	5	1.3 M	240
			KYBER512	1	800	768
**KYBER** [33]	finalist	lattice	KYBER768	3	1184	1088
			KYBER1024	5	1568	1568
			NTRUHPS2048509	1	699	699
**NTRU** [34]	finalist	lattice	NTRUHPS2048677	3	930	930
			NTRUHRSS701	3	1138	1138
			NTRUHPS4096821	5	1230	1230
			LIGHTSABER	1	672	736
**SABER** [35]	finalist	lattice	SABER	3	992	1088
			FIRESABER	5	1312	1472
			FrodoKEM640	1	9616	9720
**FrodoKEM** [27]	alternate	lattice	FrodoKEM976	3	15,632	15,744
			FrodoKEM1344	5	21,520	21,632
			ntrulpr653	1	897	1025
			ntrulpr761	2	1039	1167
**NTRUPrime** [36]	alternate	lattice	ntrulpr857	3	1184	1312
			ntrulpr953	4	1349	1477
			ntrulpr1013	4	1455	1583
			ntrulpr1277	5	1847	1975
			sntrup653	1	994	897
			sntrup761	2	1158	1039
			sntrup857	3	1322	1184
			sntrup953	4	1505	1349
			sntrup1013	4	1623	1455
			sntrup1277	5	2067	1847
			HQC128	1	2289	4481
**HQC** [37]	alternate	code	HQC192	3	4522	9026
			HQC256	5	7245	14,469
**BIKE** [38]	alternate	code	BIKE_L1	1	1541	1573
			BIKE_L3	3	3083	3115
			SIKE_P434	1	346	330
**SIKE** [39]	alternate	isogeny	SIKE_P503	2	378	402
			SIKE_P610	3	462	486
			SIKE_P751	5	564	596
**Conventional**	(None)	ECDHE	sec256r1	1	65	65
		RSA	RSA-2048	1	256	256

## Data Availability

Not applicable.

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
