# Peer review of "Secure IoT in the Era of Quantum Computers—Where Are the Bottlenecks?"

_sensors, 2022, doi:10.3390/s22072484_

Round 1

Reviewer 1 Report

This paper is entitled “Secure IoT in the Era of Quantum Computers - Where are the Bottlenecks?” and presents very important aspects related to state-of-the-art in this field.

The structure of the paper, all the state-of-the-art and background related work are very good. In terms of English writing, I think it is excellent, nothing to point out. The conducted experiments achieved coherent and innovative results and have demonstrated that it is possible to efficiently implement post-quantum safe TLS with off-the-shelf hardware on IoT edge devices.

Therefore, I fully agree with its publication in Sensors.

Author Response

Dear Reviewer #1,
We want to thank you very much for the time you have dedicated in reading our submission. We 
strongly appreciate your feedback and are highly grateful for your very good rating of our paper.

Reviewer 2 Report

An interesting article about post-quantum cryptographic uses in IoT applications. It is a challenging subject as postquantum algorithms are heavy in computations, and IoT devices are limited in many aspects as power, resources, and battery lifetime. The way the current manuscript is confusing. For instance, the authors said we could show that ..... I am not sure they did the work or can do it. Also, I am not sure how authors evaluated these schemes; most of the information in the current manuscript can be collected from public information and detailed reading. I recommend the current manuscript be rewritten to be more transparent. 

Reviewer 3 Report

The authors are providing experimental results on the use of quantum cryptography for IoT devices and reporting performance aspects of such a relevant setup. The paper is well written and easy to follow, but some concepts and results should be further clarified.

What do you mean by the term "common belief" used in the paper for citing the need for hardware accelerators? Are there experimental results from previous papers? If yes, why the contradictory outcome (i.e., no need of accelerators) of this work compared to previous ones (and which ones)? 

Why the authors are experimenting with TLS 1.2 which was updated to 1.3 not only for security but also for performance reasons? Concluding to bandwidth bottleneck issues may be attributed to the old version of TLS used in the experiments. This should be discussed. 

It seems that there are published papers that are not included in the related work. Did the authors searched thoroughly the related work??

Malina, L., Popelova, L., Dzurenda, P., Hajny, J., & Martinasek, Z. (2018). On feasibility of post-quantum cryptography on small devices. IFAC-PapersOnLine51(6), 462-467.

The authors do not conclude to a cryptographic structure? what is the smart combination of post quantum DSA that you mention in the introduction? 

Reviewer 4 Report

The authors focus their study on the field of post-quantum cryptographic algorithms by investigating the  application of  potential post-quantum Key-Encapsulation Mechanisms and Digital Signature Algorithms proposed in the related US NIST process to a state-of-the-art TLS-based low-power IoT infrastructure.

The manuscript is overall well written and easy to follow and the authors have well thought out their main contributions. The provided analysis is concrete, complete, and correct and the authors  have provided all the intermediate steps in order to enable the reader to easily follow it.

The authors should consider the following suggestions provided by the reviewer in order to improve the scientific depth of their manuscript, as well as they should address the following comments in order to improve the quality of presentation of their manuscript.

Initially, in Section 1, the authors should discuss hardware-based security solutions stemming from the implementation of the physical unclonable functions, such as Artificially Intelligent Electronic Money, doi: 10.1109/MCE.2020.3024512, by introducing zero trust security solutions of low computational complexity. The authors should make at least a qualitative discussion of the difference of the proposed approach compared to the existing state of the art.

Furthermore, it will be beneficial for the paper if the authors can provide some quantifiable numerical results in order to demonstrate the drawbacks and benefits of the proposed post-quantum Key-Encapsulation Mechanisms solution.

Finally, the manuscript has several syntax errors that the authors need to address in order to improve its readability for the average reader.

Round 2

Reviewer 3 Report

The authors have addressed the comments. 

Reviewer 4 Report

The authors have addressed in detail the reviewers’ comments.